# The Role of Vitamin D_3_ as an Independent Predicting Marker for One-Year Mortality in Patients with Acute Heart Failure

**DOI:** 10.3390/jcm11102733

**Published:** 2022-05-12

**Authors:** Kirsten Thiele, Anne Cornelissen, Roberta Florescu, Kinan Kneizeh, Vincent Matthias Brandenburg, Klaus Witte, Nikolaus Marx, Alexander Schuh, Robert Stöhr

**Affiliations:** 1Department of Internal Medicine I, University Hospital Aachen, RWTH Aachen University, 52074 Aachen, Germany; acornelissen@ukaachen.de (A.C.); rflorescu@ukaachen.de (R.F.); kkneizeh@ukaachen.de (K.K.); kwitte@ukaachen.de (K.W.); nmarx@ukaachen.de (N.M.); aschuh@ukaachen.de (A.S.); rstoehr@ukaachen.de (R.S.); 2Department of Cardiology and Nephrology, Rhein-Maas Klinikum, 52146 Wuerselen, Germany; vincent.brandenburg@rheinmaasklinikum.de; 3Department of Internal Medicine I, St. Katharinen Hospital Frechen, 50226 Frechen, Germany

**Keywords:** vitamin D_3_, acute heart failure, 1-year survival, Seattle Heart Failure Model

## Abstract

Background: Deficiency in vitamin D_3_ and its metabolites has been linked to dismal outcomes in patients with chronic diseases, including cardiovascular disease and heart failure (HF). It remains unclear if a vitamin D_3_ status is a prognostic feature in patients with acute decompensated HF. Methods: We assessed serum levels of 25-OH-vitamin D_3_ and 1,25-(OH)_2_-vitamin D_3_ in 139 patients with acute HF who had been admitted to the intermediate care unit of a maximum care hospital. The follow-up period was one year. After exclusion of patients with sampling errors and those who were lost to follow-up, 118 patients remained in the final study cohort. Outcome estimates by 25-OH-vitamin D_3_ and 1,25-(OH)_2_-vitamin D_3_ levels were compared to the Seattle Heart Failure (SHF) Model. Results: More than two-thirds (79.7%) of the patients showed inadequate 25-OH-vitamin D_3_ levels (i.e., <30 ng/mL) upon admission. Low levels of 1,25-(OH)_2_-vitamin D_3_ (i.e., <19.9 pg/mL) were observed in 16.1% of patients. Of the 118 HF patients, 22 (19%) died during the following 12 months. There were no differences in vitamin D_3_ levels between patients who died and those who survived, neither in 25-OH-vitamin D_3_ (23.37 ± 19.14 ng/mL vs. 19.11 ± 12.25 ng/mL; *p* = 0.19) nor in 1,25-(OH)_2_-vitamin D_3_ levels (31.10 ± 19.75 ng/mL vs. 38.25 ± 15.73 ng/mL; *p* = 0.02); therefore, vitamin D_3_ levels alone did not predict one-year survival (AUC [25-OH-vitamin D_3_] 0.50; 95% CI 0.34–0.65; AUC [1,25-(OH)2-vitamin D_3_] 0.62; 95% CI 0.48–0.76). Moreover, whilst the SHF model exhibited acceptable discriminatory ability for predicting one-year mortality (AUC 0.79; 95% CI 0.66–0.91), adding vitamin D levels on admission to the SHF score did not improve its discriminatory value. Conclusion: Our data do not support the use of vitamin D_3_ screening in patients admitted with acute decompensated HF to aid prognostication.

## 1. Background

Vitamin D_3_ plays a key role in calcium and phosphate homeostasis and is crucial for skeletal health. Vitamin D_3_ deficiency or insufficiency is a highly prevalent but often underestimated condition in the general population, especially among the elderly [1,2]. Over the last two decades, several studies have demonstrated a negative impact of vitamin D_3_ deficiency on morbidity and mortality in various patient populations with chronic illnesses [3,4,5,6], and associations between vitamin D_3_ deficiency and cardiovascular diseases have been reported (reviewed in [7,8]). By targeting the vitamin D_3_ receptor expressed on cardiomyocytes, vitamin D_3_ has been found to affect myocardial metabolism with beneficial effects on myocardial dysfunction and cardiac hypertrophy [9,10]. Vice versa, low levels of vitamin D_3_ and its metabolites have been associated with the development and progression of chronic heart failure (HF) [11,12,13,14], as well as with increased hospitalization and mortality in these patients [13,15,16]. Less is known about the role of vitamin D_3_ in critically ill patients with acute HF. In particular, it is unclear whether the assessment of vitamin D_3_ levels in patients with acute HF offers additional prognostic information over traditional risk factors and well-established risk estimation tools.

The aim of our project was, therefore, to investigate whether vitamin D_3_ levels, assessed at hospital admission, are associated with one-year survival in patients with acute decompensated HF. We compared outcome estimates by serum levels of 25-OH-vitamin D_3_ and 1,25-(OH)_2_-vitamin D_3_ to those obtained from the Seattle Heart Failure (SHF) Model as an established prognostic model of HF survival [17].

## 2. Methods

### 2.1. Study Population and Study Design

From August 2016 to September 2018, 139 unselected, consecutive patients with acute HF who had been admitted to the intermediate care unit of University Hospital RWTH Aachen were included in the study. Inclusion criteria were signs and symptoms of acute HF associated with elevated NT-proBNP ≥ 300 pg/mL in patients above 18 years of age. Exclusion criteria were age < 18 years, admission or transfer to the Intensive Care Unit within ≤12 h, CKD requiring dialysis, heart or kidney transplantation, and pregnancy or breastfeeding. Patients who died within 12 h after admission were also excluded. Patients were screened during business hours of the clinical study center from Mondays–Thursdays between 7 a.m. and 1 p.m. The study protocol was approved by the Independent Ethics Committee at the RWTH Aachen Faculty of Medicine (EK 099/16). All methods were in accordance with current data protection guidelines and the Declaration of Helsinki. All patients provided written informed consent before study enrollment.

### 2.2. Definition of Acute HF

Acute HF was defined pursuant to current ESC guidelines [18] as the rapid or gradual onset of typical clinical symptoms and signs of congestion (e.g., dyspnea, peripheral and/or pulmonary edema, and/or signs of increased central venous pressure), along with evidence of structural and/or functional cardiac abnormalities and/or elevated NT-proBNP ≥ 300 pg/mL. Patients with acute new-onset HF and those with acutely decompensated chronic HF were both defined as “acute HF”. All patients underwent echocardiography within the first 72 h after hospitalization. A left-ventricular ejection fraction (LVEF) of ≥50% in a patient with acute HF was classified as HF with preserved LVEF (HFpEF). Acute HF with an LVEF between 41 and 49% was defined as HF with mildly reduced LVEF (HfmrEF), and acute HF with an LVEF ≤ 40% was categorized as HF with reduced EF (HfrEF). NYHA and Killip functional classifications were utilized to classify the severity of symptoms.

### 2.3. Blood Sampling

Blood was collected in all patients at the time of admission. Routine laboratory analyses, including serum levels of 25-OH-vitamin D_3_ and 1,25-(OH)_2_-vitamin D_3_, were performed at our core laboratory facility.

### 2.4. Study Endpoints and Follow-Up

This retrospective analysis aimed to identify any association between 25-OH-vitamin D_3_ and 1,25-(OH)_2_-vitamin D_3_ and one-year survival in acute HF patients. Secondary analysis examined these metabolites as biomarkers for prognosis estimation compared with the SHF Model. In addition, we investigated whether 25-OH-vitamin D_3_ and 1,25-(OH)_2_-vitamin D_3_ were associated with HF severity as assessed by clinical and echocardiographic parameters. All patients were followed up via phone calls to assess vital status at one year after hospital admission.

### 2.5. Risk Calculation

The SHF Model [17] was utilized to estimate the anticipated one-year survival (https://qxmd.com/calculate/calculator_203/seattle-heart-failure-model; 16 December 2020). Parameters that were entered into the calculation included age, sex, weight, systolic blood pressure, NYHA class, etiology (ischemic vs. non-ischemic), LVEF, diuretic dosages, usage of allopurinol, statins, ACE inhibitors or angiotensin-receptor blockers, beta-blockers, the presence of implantable cardioverter defibrillators and/or cardiac resynchronization therapy, and blood levels of sodium, hemoglobin, lymphocytes, cholesterol, and uric acid.

### 2.6. Statistical Analysis

Standard descriptive statistics were used to assess the baseline characteristics of the cohort. Categorical data are expressed as numbers and percentages; continuous data are presented as mean ± SD. Normality was assessed using the Shapiro–Wilk test. Serum levels of 25-OH-vitamin D_3_ and 1,25-(OH)_2_-vitamin D_3_ were not normally distributed and, therefore, log transformed (log25-OH-vitamin D_3_ and log1,25-(OH)_2_-vitamin D_3_). The ROUT method [19] was used to identify potential outliers based on a false discovery rate of 1%. Statistical differences between categorical variables were determined using a chi-square test. Student’s t-test or one-way ANOVA and Tukey’s multiple comparisons test were used to find differences between continuous variables. We performed direct logistic regression analyses to evaluate associations between log25-OH-vitamin D_3_/log1,25-(OH)_2_-vitamin D_3_ and one-year survival. Each model was adjusted for age, sex, eGFR, and vitamin D supplementation. In addition, areas under the ROC curve (AUCs) and 95% confidence intervals (CI) were calculated to compare the predictive abilities of the SHF Model and vitamin D_3_ with respect to one-year mortality. Graphs were created in GraphPad Prism (GraphPad Software, Version 8.4.1). Statistical analyses were performed using IBM SPSS Statistics, version 28.0.

## 3. Results

### 3.1. Baseline Characteristics

The baseline characteristics of the population were previously described [20]. In short, 139 patients with acute HF who were admitted to the Intermediate Care Unit of University Hospital Aachen were included in the initial study. Baseline vitamin D status and one-year follow up data were available for 118 patients (Figure 1). Baseline characteristics are presented in Table 1. The mean age of study participants was 66.6 (range 32.0–94.0), and 74 patients (71.2%) were male. In 34 patients (28.8%), HF was previously known (chronic decompensated HF), whereas 84 patients (71.2%) were diagnosed with de novo HF. The mean LVEF was 38.9 ± 10.5%. Ten patients (8.5%) were classified as HfpEF, forty-nine patients (41.5%) were classified as HfmrEF, and fifty-nine patients (50.0%) were classified as HfrEF. The mean eGFR was 82.95 ± 40.65 mL/min/1.73 m^2^. Of the study population, 66.1% (78 patients) showed no or only mild kidney damage (CKD I or II), 27.1% (32 patients) had a moderate impairment of kidney function (CKD III), and 6.8% (8 patients) showed severe CKD. Over the course of the one-year follow-up period, 22 patients (18.6%) died.

### 3.2. Levels of 25-OH-Vitamin D_3_ and 1,25-(OH)_2_-Vitamin D_3_ in Patients with HF

Serum levels of 25-OH-vitamin D_3_ and 1,25-(OH)_2_-vitamin D_3_ were assessed at hospital admission in all patients (Table 1). Mean 25-OH-vitamin D_3_ was 19.90 ± 13.80 ng/mL. Ninety-four patients (79.7%) had inadequate 25-OH-vitamin D_3_ levels (i.e., ≤30 ng/mL) (Table 2). Patients with 25-OH-vitamin D_3_ inadequacy had significantly higher eGFR compared with patients with 25-OH-vitamin D_3_ adequacy in our study. We did not observe any differences in LVEF, LVEDD, LVESD, LA area, or NT-proBNP levels between those with 25-OH-vitamin D_3_ inadequacy and those with adequate levels of 25-OH-vitamin D_3_. There were also no differences in the clinical severity of HF upon admission, as assessed by NYHA status.

Nineteen patients (16.1%) had low 1,25-(OH)_2_-vitamin D_3_ levels (i.e., <19.9 pg/mL) at baseline (Table 3). While we did not observe differences in LVEF between patients who had low 1,25-(OH)_2_-vitamin D_3_ and those who had normal levels, patients with low 1,25-(OH)_2_-vitamin D_3_ had significantly higher NT-proBNP, as well as larger LVEDD, LVESD, and LA areas. Patients with low 1,25-(OH)_2_-vitamin D_3_ also had a lower eGFR and lower anticipated survival at one year, as assessed by the SHF model.

### 3.3. Associations between Levels of Vitamin D_3_ and One-Year Mortality

Twenty-two patients (18.6%) had died at one-year follow-up. We observed no differences in 25-OH-vitamin D_3_ levels between patients who died and those who survived (23.37 ± 19.14 ng/mL vs. 19.11 ± 12.25 ng/mL; *p* = 0.69), while 1,25-(OH)_2_-vitamin D_3_ was significantly lower in patients who died compared to those who survived (31.10 ± 19.75 pg/mL vs. 38.25 ± 15.73 pg/mL; *p* = 0.009). Likewise, 1,25-(OH)_2_-vitamin D_3_ was inversely associated with one-year mortality in an unadjusted logistic regression (OR 0.11; 95% CI 0.02–0.64; *p* = 0.01) (Figure 2a). However, the significance was lost after adjusting for age, sex, eGFR, and vitamin D supplementation (OR 0.35; 95% CI 0.05–2.49; *p* = 0.30; Figure 2b). No association was observed between log25-OH-vitamin D_3_ and death at one year (unadjusted OR 1.40; 95% CI 0.26–7.45; *p* = 0.69; adjusted OR 0.80; 95% CI 0.11–5.89; *p* = 0.82; Figure 2c).

We calculated AUCs to assess the predictive ability of 1,25-(OH)_2_-vitamin D_3_ and 25-OH-vitamin D_3_ in comparison with the SHF model. While SHF had an acceptable discriminatory ability to distinguish between survivors and non-survivors after the one-year follow-up period (AUC 0.79; 95% CI 0.66–0.91) and correctly classified survival at one year in 85.6% of patients, neither 1,25-(OH)_2_-vitamin D_3_ (AUC 0.62; 95% CI 0.48–0.76) nor 25-OH-vitamin D_3_ (AUC 0.50; 95% CI 0.34–0.65) predicted one-year survival with acceptable accuracy (Figure 3). The addition of vitamin D levels on admission to the SHF score did not improve its discriminatory value, with 85.6% patients who were correctly classified.

## 4. Discussion

In this single-center study including 118 patients with acute HF admitted to the intermediate care unit of a maximum care hospital, deficient or inadequate serum levels of 25-OH-vitamin D_3_ and 1,25-(OH)_2_-vitamin D_3_ were not predictive for mortality at one year. In our study collective, only 20.3% of patients had adequate serum levels of 25-OH-vitamin D_3_, whereas 79.7% had inadequate levels of 25-OH-vitamin D_3_. While this tremendously high prevalence of hypovitaminosis D among critically ill patients with acute HF might support the hypothesis that low levels of vitamin D_3_ are associated with the occurrence of cardiac events and a worse prognosis, data from the general population in Germany indicate that between 29% and 33% of all adults have 25-OH-vitamin D_3_ serum levels below 12 ng/mL (30 nmol/L), with even higher numbers among the elderly [21,22,23].

Deficiency in vitamin D_3_ and its metabolites is known to be linked to the development and the progression of chronic HF [11,12,14]. Over the last few decades, several studies have sought to elucidate the underlying molecular mechanisms by which vitamin D_3_ may exert its beneficial effects in patients with cardiovascular diseases [9,10,24,25,26]. Vitamin D deficiency, as well as the deletion of the vitamin D receptor, resulted in increased levels of renin and angiotensin II in mice, eventually leading to hypertension and cardiac hypertrophy [9,27], whereas the injection of 1,25-(OH)_2_-vitamin D_3_ suppressed renin synthesis [28]. Furthermore, vitamin D_3_ has a role in the regulation of immunomodulatory and inflammatory processes by affecting T helper cells and inflammatory cytokines [25,29,30].

Several studies have suggested associations between low vitamin D_3_ and an increased mortality risk in patients with cardiovascular diseases and HF [6,11,12,13,31]. Data from the LURIC (Ludwigshafen Risk and Cardiovascular Health) study in 1801 patients with cardiovascular disease and metabolic syndrome showed significantly lower all-cause and cardiovascular mortality in 143 patients with optimal levels of 25-OH-vitamin D_3_ (i.e., ≥75 nmol/L = ≥30 ng/mL) in comparison to 487 patients with severe vitamin D_3_ deficiency (i.e., <25 nmol/L = <10 ng/mL) [6]. These data are consistent with a study by Cubbon et al. who also found increased mortality in 1802 ambulatory patients with chronic HF and vitamin D_3_ deficiency [32]. In line with this, Liu et al. reported an independent association between low 25-OH-vitamin D_3_ levels and all-cause mortality in 548 patients hospitalized for mild-to-moderate HF [13]. Furthermore, recently published meta-analysis summarizing data from 13 clinical trials that enrolled a total of 1215 patients suggested an improved ejection fraction with less adverse remodeling by vitamin D supplementation [33].

However, while previous studies mostly investigated associations with mortality in chronic HF, less is known about the role of vitamin D in critically ill patients with acute HF. Data from 80 patients with acute deterioration of a chronic illness, 20 of whom had acute decompensated HF, suggested an inverse correlation between 25-OH-vitamin D_3_ and the length of hospital stay, as well as significant differences in vitamin D levels between survivors and non-survivors during the hospitalization period [34]. Another study including 135 intensive care patients, 26 of whom had concomitant HF, found an increased mortality risk over the course of the 28-day observation period in patients with 25-OH-vitamin D_3_ serum levels below 12 ng/mL [35]. However, to the best of our knowledge, no study has yet investigated the ability of vitamin D_3_ to predict mortality at one year in a critically ill collective of patients with acute HF.

In HF patients, risk stratification is essential for clinical management. In addition to well-established models of projected survival, such as the SHF model, aiming to standardize prognosis estimation in clinical practice, reliable prognostic biomarkers could potentially facilitate prognosis estimation and, thus, allow treatment stratification. We have recently shown that elevated levels of fibroblast growth factor 23 (FGF23), secreted in response to elevated phosphate levels by osteoblasts and osteocytes, predicted one-year mortality with similar accuracy as the SHF model in patients with acute HF [20]. Interestingly, FGF23 is a potent suppressor of vitamin D, inhibiting the conversion of 25-OH-vitamin D_3_ to its active form, 1,25-(OH)_2_-vitamin D_3_. While there were no differences in 25-OH-vitamin D_3_ levels between survivors and non-survivors in our study, we observed higher 1,25-(OH)_2_-vitamin D_3_ levels among survivors of the one-year follow-up period. However, significance was lost after ajusting for age, sex, and eGFR. Thus, our data do not support an independent effect of low vitamin D_3_ on mortality either in its inactive or active form. One could infer 1,25-(OH)_2_-vitamin D_3_ to be a mere bystander, reflecting the negative effects of impaired kidney function and/or elevated FGF23 levels rather than having an independent causal role in HF outcome. There were also no differences in 25-OH-vitamin D_3_ levels between survivors and non-survivors of the one-year follow-up period. However, we noted a higher eGFR in patients with 25-OH-vitamin D_3_ deficiency, which is most likely coincidental. As declining eGFR is well-known to dramatically increase mortality in HF patients [36], higher eGFR in patients with 25-OH-vitamin D_3_ deficiency might have confounded our results. Our main finding that 25-OH-vitamin D_3_ deficiency is not relevant to aid in prognostication in patients with acute HF is in contrast to several previous studies that have highly praised vitamin D_3_ for predicting HF-related outcomes, concluding that vitamin D_3_ supplementation may reduce mortality [37]. Along with our findings, however, more recent work has challenged the significance of vitamin D supplementation regarding mortality in patients with HF or cardiovascular disease. In a study with 400 HF patients with vitamin D_3_ deficiency who received 4000 IE vitamin D daily or placebo for 3 years, no difference in mortality was observed between the treatment and the control group [38]. A recent single-center study including 189 patients hospitalized for acute HF showed that intensified efforts in screening and treatment of hypovitaminosis D did not improve outcome at 6 months [39]. Likewise, early administrations of high-dose vitamin D_3_ showed no advantage over placebo with respect to 90-day mortality in a collective of 1360 critically ill patients with 25-OH-vitamin D_3_ serum levels below 20 ng/mL in a recent multicenter study [40]. Finally, Mendelian randomization studies failed to demonstrate associations between genetic variants affecting vitamin D_3_ levels and cardiovascular mortality, suggesting that hypovitaminosis D is associated with but not necessarily causal for higher cardiovascular mortality [41].

Our study has several limitations. Importantly, our study cohort was comparatively small, and the generalizability of our findings is limited by the single-center design of our study. Another relevant limitation is the short follow-up time of only one year. Thus, long-term effects of vitamin D supplementation on cardiovascular outcome in patients with acute HF remain uncertain. In addition, we did not consider gender-specific differences in calcium homeostasis as our study cohort was mostly composed of men. However, the greater percentage of men compared to women in our study might be explained by a generally higher incidence and prevalence of HF in men [42]. Furthermore, patients with higher eGFR tended to exhibit lower 25-OH-vitamin D_3_ levels, which is most likely coincidental. As declining eGFR is well known to dramatically increase mortality in HF patients [36], higher eGFR in patients with 25-OH-vitamin D_3_ deficiency might have confounded our results. Finally, we did not account for seasonal differences in vitamin D assessment in our study. Larger multicentric studies are warranted to confirm that vitamin D alone has no ability to independently predict outcome in critically ill patients with acute HF.

## 5. Conclusions

Although the small sample size of our study limits the generalizability of the findings, our data could bolster the hypothesis that deficiency of vitamin D_3_ is a phenomenon that is frequently seen in critically ill patients and often correlates with severity of disease but do not support the use of vitamin D_3_ status as a prognostic feature in patients admitted with acute decompensated HF. Neither 25-OH-vitamin D_3_ nor 1,25-(OH)_2_-vitamin D_3_ were able to predict mortality at one year with sufficient accuracy in our study compared to the SHF Model.

## Figures and Tables

**Figure 1 jcm-11-02733-f001:**
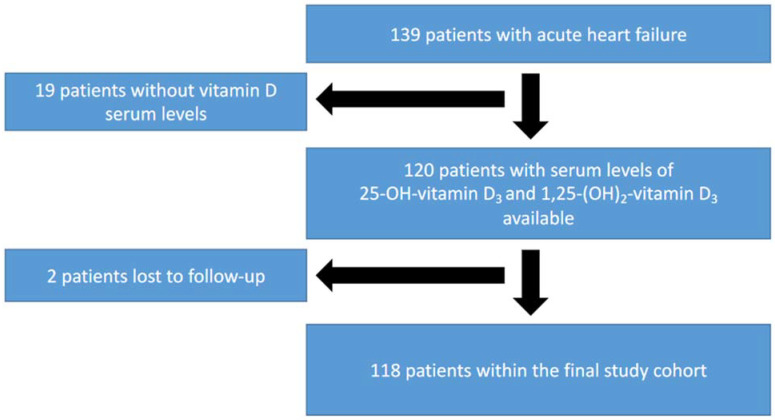
Study cohort. From 139 patients hospitalized for acute HF, 1,25-(OH)_2_-vitamin D_3_ and 25-OH-vitamin D_3_ levels were available in 120 patients. Two patients were lost to follow-up. The final study cohort comprised 118 patients.

**Figure 2 jcm-11-02733-f002:**
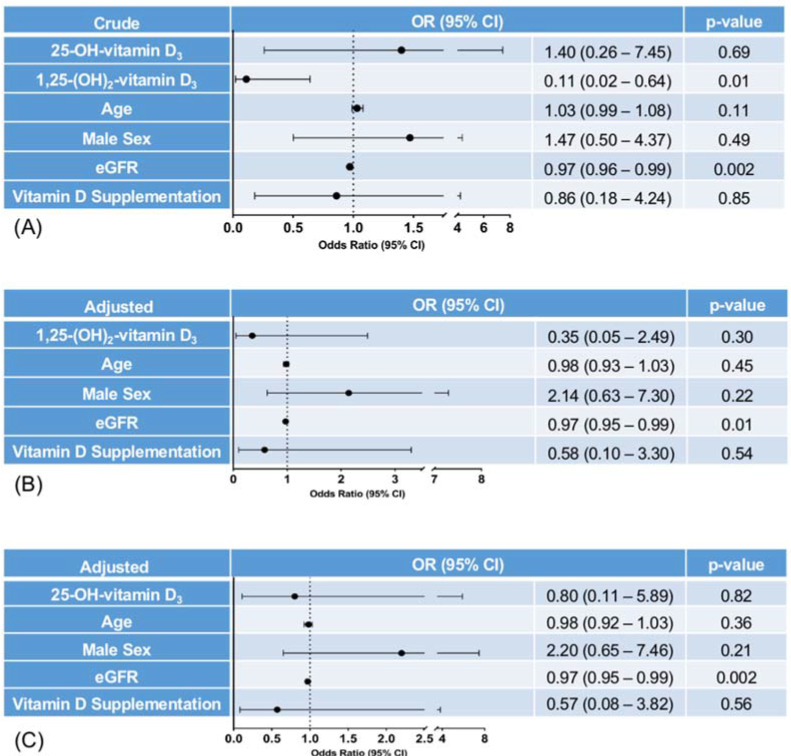
Logistic regression analyses assessing associations between vitamin D levels and death at one year. (**A**) In the crude model, 1,25-(OH)_2_-vitamin D_3_ levels were inversely associated with death at one year, while there was no such association with 25-OH-vitamin D_3_. Likewise, a decrease in eGFR was associated with higher odds of death at one year. (**B**) After adjustment for age, male sex, eGFR, and vitamin D supplementation, there was no significant association between 1,25-(OH)_2_-vitamin D_3_ levels and death at one year anymore. (**C**) Likewise, 25-OH-vitamin D_3_ levels were not associated with death at one year after adjustment for covariates.

**Figure 3 jcm-11-02733-f003:**
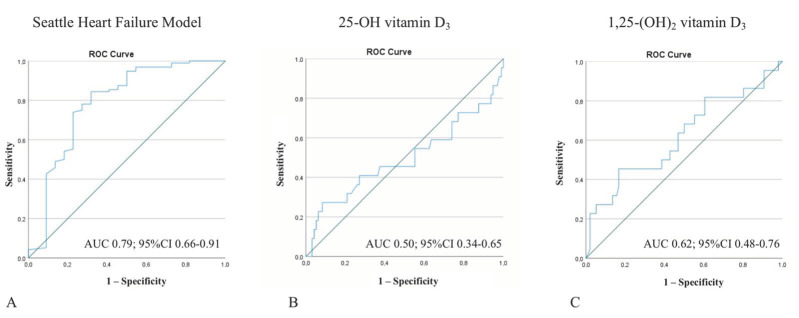
Receiver-operating curves for the SHF model, 25-OH-vitamin D_3_, and 1,25-(OH)_2_-vitamin D_3_. While the SHF model exhibited acceptable discriminatory ability to distinguish between survivors and non-survivors of the one-year follow-up period (**A**), neither 25-OH-vitamin D_3_ (**B**) nor 1,25-(OH)_2_-vitamin D_3_ (**C**) predicted one-year survival with acceptable accuracy.

**Table 1 jcm-11-02733-t001:** Baseline characteristics.

	Study Cohort (*n* = 118)
Age, years (Range)	66.58 (32–94)
Male gender, *n* (%)	84 (71.2%)
BMI, kg/m^2^	27.57 ± 5.32
Systolic blood pressure, mmHg	119.42 ± 24.75
De novo Heart Failure, *n* (%)	84 (71.2%)
Chronic Heart Failure, *n* (%)	34 (28.8%)
Etiology of Heart Failure	
Ischemic cardiomyopathy, *n* (%)	93 (78.8%)
Non ischemic cardiomyopathy, *n* (%)	25 (21.2%)
LVEF	
≥50%, *n* (%)	10 (8.5%)
41–49%, *n* (%)	49 (41.5%)
≤40%, *n* (%)	59 (50.0%)
LVEF mean ± SD (%)	38.92 ± 10.51
NYHA classification	
1	0 (0.0%)
2	15 (12.7%)
3	58 (49.2%)
4	45 (38.1%)
Medication	
Beta blockers, *n* (%)	108 (91.5%)
ACE inhibitors/Angiotensin II receptor blockers, *n* (%)	108 (91.5%)
Loop Diuretics, *n* (%)	72 (61.0%)
Potassium Sparing Diuretics, *n* (%)	37 (31.4%)
Risk Factors	
Hypertension, *n* (%)	69 (58.5%)
Smoking, *n* (%)	66 (55.9%)
Diabetes, *n* (%)	39 (33.1%)
Hypercholesterolemia, *n* (%)	68 (57.6%)
Chronic kidney disease, *n* (%)	
CKD Level 1, *n* (%)	45 (38.1%)
CKD Level 2, *n* (%)	33 (28.0%)
CKD Level 3, *n* (%)	32 (27.1%)
CKD Level 4, *n* (%)	8 (6.8%)
CKD Level 5, *n* (%)	0 (0.0%)
Blood parameters	
Estimated GFR, mL/min/1.73 m^2^	82.95 ± 40.65
NT-proBNP, pg/mL	5747 ± 7227
Calcium, mmol/L	2.18 ± 0.15
Phosphorus, mmol/L	1.08 ± 0.29
Parathormone, pg/mL	36.50 ± 29.64
1,25-(OH)_2_-vitamin D_3_, pg/mL	36.92 ± 16.70
25-OH-vitamin D_3_, ng/mL	19.90 ± 13.80
Low 1,25-(OH)_2_-vitamin D_3_ (<19.9 pg/mL), *n* (%)	19 (16.1%)
25-OH-vitamin D_3_	
Inadequacy (10–30 ng/mL)	94 (79.7%)
Adequacy (>30 ng/mL)	24 (20.3%)
Vitamin D Supplementation	12 (10.2%)
Risk Estimation	
Anticipated 1-year survival (Seattle Heart Failure Model), %	84.72 ± 24.22
Survival 1 year, *n* (%)	96 (81.4)

BMI = body mass index, LVEF = left ventricular ejection fraction, SD = standard deviation, GFR = glomerular filtration rate, NT-proBNP = N-terminal pro B-type natriuretic peptide.

**Table 2 jcm-11-02733-t002:** 25-OH-vitamin D_3_ levels in HF patients (*n* = 118).

25-OH-VitD_3_	Inadequacy (≤30 ng/mL) *n* = 94	Adequacy (>30 ng/mL) *n* = 24	*p*-Value
Age	66.50 ± 13.09	66.56 ± 10.94	0.89
Male Sex	67 (71.3%)	17 (70.8%)	0.97
Death	16 (17.0%)	6 (25.0%)	0.37
De novo Heart Failure	72 (76.6%)	12 (50.0%)	0.01
Chronic Heart Failure	22 (23.4%)	12 (50.0%)	0.01
NYHA Classification			
I	0 (0.0%)	0 (0.0%)	NA
II	12 (12.8%)	3 (12.5%)	0.97
III	45 (47.9%)	13 (54.2%)	0.58
IV	37 (39.4%)	8 (33.3%)	0.59
Echocardiography			
LVEF			
≥50%, *n* (%)	8 (8.5%)	2 (8.3%)	0.98
41–49%, *n* (%)	43 (45.7%)	6 (25.0%)	0.07
≤40%, *n* (%)	43 (45.7%)	16 (66.7%)	0.07
LVEF mean ± SD (%)	39.33 ± 10.68	37.33 ± 9.86	0.41
LVEDD (mm)	52.16 ± 7.85	51.82 ± 7.29	0.87
LVESD (mm)	40.40 ± 10.19	40.53 ± 9.94	0.96
LA area (mm^2^)	21.81 ± 5.94	21.44 ± 5.66	0.82
TAPSE (cm)	2.00 ± 0.47	2.19 ± 0.53	0.15
RVSP (mmHg)	31.93 ± 11.88	30.94 ± 10.88	0.77
Blood parameters			
NT-proBNP, pg/mL	5528.28 ± 7284.81	6569.40 ± 7094.48	0.53
Estimated GFR, mL/min/1.73 m^2^	86.94 ± 41.42	67.29 ± 33.85	0.03
Calcium, mmol/L	2.17 ± 0.16	2.19 ± 0.11	0.53
Phosphorus, mmol/L	1.06 ± 0.27	1.15 ± 0.36	0.20
Parathormone, pg/mL	37.58 ± 30.77	32.00 ± 24.43	0.43
1,25-(OH)_2_-vitamin D_3_, pg/mL, day 1	36.20 ± 16.45	39.74 ± 17.71	0.36
25-OH-vitamin D_3_, ng/mL, day 1	13.96 ± 6.20	43.16 ± 10.21	<0.001
Vitamin D Supplementation, *n* (%)	1 (1.1%)	11 (45.8%)	<0.001
Risk Estimation			
Anticipated 1-year survival (Seattle Heart Failure Model)	87.22 ± 21.22	74.93 ± 32.19	0.03

LVEF = left ventricular ejection fraction, SD = standard deviation, LVEDD = left ventricular end-diastolic diameter, LVESD = left ventricular end-systolic diameter, LA = left atrium, NT-proBNP = N-terminal pro B-type natriuretic peptide, GFR = glomerular filtration rate.

**Table 3 jcm-11-02733-t003:** 1,25-OH-vitamin D_3_ levels in HF patients (*n* = 118).

1,25-(OH)_2_-Vitamin D_3_	Deficiency (<19.9 pg/mL) *n* = 19	Adequacy (>19.9 pg/mL) *n* = 99	*p*-Value
Age	65.59 ± 12.80	71.79 ± 10.55	0.05
Male Sex	13 (68.4%)	71 (71.7%)	0.77
Death	6 (31.6%)	16 (16.2%)	0.11
De novo Heart Failure	11 (57.9%)	73 (73.7%)	0.16
Chronic Heart Failure	8 (42.1%)	26 (26.3%)	0.16
NYHA Classification			
I	0 (0.0%)	0 (0.0%)	NA
II	1 (5.3%)	14 (14.1%)	0.29
III	7 (36.8%)	51 (51.5%)	0.24
IV	11 (57.9%)	34 (34.3%)	0.05
Echocardiography			
LVEF			
≥50%, *n* (%)	5 (26.3%)	5 (5.1%)	0.002
41–49%, *n* (%)	3 (15.8%)	46 (46.5%)	0.01
≤40%, *n* (%)	11 (57.9%)	48 (48.5%)	0.45
LVEF mean ± SD (%)	38.21 ± 15.29	39.06 ± 9.42	0.75
LVEDD (mm)	56.00 ± 8.28	51.44 ± 7.47	0.05
LVESD (mm)	45.85 ± 11.13	39.50 ± 9.67	0.04
LA area (mm^2^)	25.85 ± 6.54	21.03 ± 5.47	0.005
TAPSE (cm)	2.17 ± 1.37	2.24 ± 1.45	0.76
RVSP (mmHg)	39.00 ± 16.82	30.70 ± 10.73	0.07
Blood parameters			
NT-proBNP, pg/mL	10,118.27 ± 8187.76	4873.30 ± 6731.15	0.003
Estimated GFR, mL/min/1.73 m^2^	55.66 ± 32.81	88.18 ± 40.03	0.001
Calcium, mmol/L	2.18 ± 0.15	2.18 ± 0.13	0.93
Phosphorus, mmol/L	1.10 ± 0.36	1.07 ± 0.28	0.68
Parathormone, pg/mL	40.13 ± 27.21	35.82 ± 30.15	0.57
1,25-(OH)_2_-vitamin D_3_, pg/mL	13.11 ± 5.53	41.49 ± 14.01	<0.001
25-OH-vitamin D_3_, ng/mL	14.75 ± 11.10	20.89 ± 14.09	0.08
Vitamin D Supplementation, *n* (%)	11 (11.1%)	1 (5.3%)	0.44
Risk Estimation			
Anticipated 1-year survival (Seattle Heart Failure Model), %	68.74 ± 37.37	87.79 ± 19.61	0.001

LVEF = left ventricular ejection fraction, SD = standard deviation, LVEDD = left ventricular end-diastolic diameter, LVESD = left ventricular end-systolic diameter, LA = left atrium, NT-proBNP = N-terminal pro B-type natriuretic peptide, GFR = glomerular filtration rate.

## Data Availability

Data are available upon reasonable request to the corresponding author.

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
