# Peer review of "The Role of Vitamin D_3_ as an Independent Predicting Marker for One-Year Mortality in Patients with Acute Heart Failure"

_jcm, 2022, doi:10.3390/jcm11102733_

Round 1
Reviewer 1 Report
Dear Authors,
Greetings. Good study, but the patient sample needs to be bigger to see any significant association between vitamin D levels and HF. Also, 1-year follow-up is too small of a time. As mentioned by the authors, both genders need to be included. Also, many pre-clinical studies have shown positive outcomes with vitamin D supplementation.
Author Response
Dear Authors,
Greetings. Good study, but the patient sample needs to be bigger to see any significant association between vitamin D levels and HF. Also, 1-year follow-up is too small of a time. As mentioned by the authors, both genders need to be included. Also, many pre-clinical studies have shown positive outcomes with vitamin D supplementation.
- We thank the reviewer for this important comment. We acknowledge these relevant limitations of our study which we have touched upon in our limitation section. We have expanded on the limitations to account for the fact that our study follow-up time was limited. We hope that despite these limitations our study may still be considered for publication.
- Page 11, paragraph 2:
Our study has several limitations. Importantly, our sample size was rather small, and the single-center design of our study limits the generalizability of our findings. Another relevant limitation is the short follow-up time of only one year thus long-term effects of vitamin D supplementation on cardiovascular outcome in patients with acute HF remain uncertain. In addition, we did not consider gender-specific differences in the calcium homeostasis as our study cohort was largely male, which, however, might be explained by the higher incidence and prevalence of HF in men that has been reported at all ages [42]. Furthermore, patients with higher eGFR tended to exhibit lower 25-OH-vitamin D3 levels which is most likely coincidental. As declining eGFR is well-known to dramatically increase mortality in HF patients [36], higher eGFR in patients with 25-OH-vitamin D3 deficiency might have confounded our results. Finally, we did not account for seasonal differences in vitamin D assessment in our study. Larger studies, ideally in a multicentric setting, are necessary to confirm vitamin D alone has no ability to independently predict outcome in critically ill patients with acute HF.
Reviewer 2 Report
The main aim of this manuscript was the assessment of vitamin D3 levels in patients with acute heart failure offers additional prognostic information over traditional risk factors and well-established risk estimation tools. There is still uncertainty whether Vitamin D supplementation plays a cardioprotective role.
Even though it is a small sample size study, the manuscript is concise, well written with the results that support the conclusion.
Minor comments:
- Please comment on the effect of vitamin D supplementation on LV ejection fraction or left ventricular remodeling in the discussion section.
- Higher eGFR in patients with 25-OH-vitamin D3 deficiency might have confounded results, thus it should be written in limitations.
- As the author stated in study limitations sample size of the study was small, and limits the generalizability of the findings. So, the conclusions of the study must be interpreted with caution and need to be retested and confirmed in future better designed studies.
- Higher eGFR in patients with 25-OH-vitamin D3 deficiency might have confounded the results that should be a limitation of the study.
Reviewer 3 Report
Dear Authors,
The work “The role of vitamin D3 as independent predicting marker for one-year mortality in patients with acute heart failure” is an important point in the discussion on the role of vitamin D in cardiovascular diseases. Some clinical trials have linked vitamin D deficiency to cardiovascular disease, but establishing a causal relationship between vitamin D and CVD is still at an initial stage. The authors attempted to answer the question: whether vitamin D (25(OH)D and/or 1.25(OH)2D) levels are predictors of mortality in patients with acute heart failure. They showed that vitamin D deficiency often occurs in critically ill patients. At the same time, they found that vitamin D levels were not a good enough marker to predict the annual mortality of patients with acute decompensated HF.
It is worth noting that most of the publications in this field concern chronic heart failure, and the authors of this manuscript focused on acute heart failure.
An interesting work, possible to quote seems to be the publication: Vernie Soh, Shawn Jia Xiang Tan, Rijuvani Sehgal, Manasi Mahesh Shirke, Amr Ashry, and Amer Harky. The Relationship Between Vitamin D Status and Cardiovascular Diseases. Curr Probl Cardiol 2021; 46: 100836. Maybe it was worth enriching the manuscript with this publication.
The research was well planned, the results subjected to detailed statistical analysis and presented in the form of legible and clear tables and figures. The final conclusion is fully justified by the presented results. The work is sufficient to be published in the Journal of Clinical Medicine.
Yours faithfully,
Reviewer 4 Report
The manuscript by Kirsten Thiele et al. entitled “The role of vitamin D3 as independent predicting marker for one-year mortality in patients with acute heart failure” aimed to investigate whether vitamin D3 levels, assessed at hospital admission, are associated with survival at one-year in patients with acute decompensated HF. Specifically, the authors compared outcome estimates by serum levels of 25-OH-vitamin D3 and 1,25-(OH)2-vitamin D3 to those obtained from the well-established Seattle Heart Failure (SHF) Model,
The article is well written and leads some evidence to such point; however, some major issues need to be addressed to improve the significance and reliability of the results of the study:
-Firstly, the exclusion and inclusion criteria are not clear.
-It would be important to report in Tables the different classes of drugs taken by patients. Were the patients all taking optimal medical therapy?
-It would be important to report echocardiographic parameters of right heart function, such as TAPSE. Please, add these parameters in Tables.
- Vitamin D (VitD) and parathyroid hormone (PTH) represent pillars in the homeostasis of calcium and bone metabolism, through their reciprocal regulation. Specifically, a close inverse relationship between VitD serum levels and PTH exists: indeed, in the case of VitD insufficiency, parathyroid gland is stimulated to release PTH. Recently, VitD insufficiency has been proposed to be linked to increased cardiovascular risk through multiple mechanisms; however, the role of VitD in predicting cardiovascular outcomes is at most controversial. Although in the general population VitD fails to predict cardiovascular risk, PTH represents a more reliable and promising biomarker, and the supplementation of VitD may become an important stronghold to reduce the CVR through the decrease of serum PTH values.
Consequently, it should be important to know the PTH values of these patients and this articles could be cited: “Vitamin D, parathyroid hormone and cardiovascular risk: the good, the bad and the ugly” PMID: 29252600 PMCID: PMC5757656 DOI: 10.2459/JCM.0000000000000614
As a result, the Reviewer suggests reconsidering the article after major revision.
Author Response
The manuscript by Kirsten Thiele et al. entitled “The role of vitamin D3 as independent predicting marker for one-year mortality in patients with acute heart failure” aimed to investigate whether vitamin D3 levels, assessed at hospital admission, are associated with survival at one-year in patients with acute decompensated HF. Specifically, the authors compared outcome estimates by serum levels of 25-OH-vitamin D3 and 1,25-(OH)2-vitamin D3 to those obtained from the well-established Seattle Heart Failure (SHF) Model.
The article is well written and leads some evidence to such point; however, some major issues need to be addressed to improve the significance and reliability of the results of the study:
- Firstly, the exclusion and inclusion criteria are not clear.
- We thank the reviewer for this comment and we have concretized the exclusion and inclusion criteria:
- Page 2, paragraph 1:
Inclusion criteria were signs and symptoms of acute HF, associated with elevated NT-proBNP ≥300pg/mL in patients above 18 years. Exclusion criteria were age <18 years, pregnancy or breastfeeding, CKD requiring dialysis, heart or kidney transplantation, and primary admission to the Intensive Care Unit or transfer to the Intensive Care Unit within 12 h after admission to the Intermediate Care Unit. Furthermore, patients who died within 12 h after admission were excluded from the study.
-It would be important to report in Tables the different classes of drugs taken by patients. Were the patients all taking optimal medical therapy?
- We have added heart failure relevant baseline medication to Table 1 (Baseline Characteristics). Above 90% of study participants received optimal medical therapy including RAAS inhibitors and betablockers.
-It would be important to report echocardiographic parameters of right heart function, such as TAPSE. Please, add these parameters in Tables.
- We have added echocardiographic parameters of right heart function (TAPSE, RVSP) to Table 2 and 3.
- Vitamin D (VitD) and parathyroid hormone (PTH) represent pillars in the homeostasis of calcium and bone metabolism, through their reciprocal regulation. Specifically, a close inverse relationship between VitD serum levels and PTH exists: indeed, in the case of VitD insufficiency, parathyroid gland is stimulated to release PTH. Recently, VitD insufficiency has been proposed to be linked to increased cardiovascular risk through multiple mechanisms; however, the role of VitD in predicting cardiovascular outcomes is at most controversial. Although in the general population VitD fails to predict cardiovascular risk, PTH represents a more reliable and promising biomarker, and the supplementation of VitD may become an important stronghold to reduce the CVR through the decrease of serum PTH values.
Consequently, it should be important to know the PTH values of these patients and this articles could be cited: “Vitamin D, parathyroid hormone and cardiovascular risk: the good, the bad and the ugly” PMID: 29252600 PMCID: PMC5757656 DOI: 10.2459/JCM.0000000000000614
- We thank the reviewer for this suggestion. We have added the PTH values to all tables. Furthermore, we have now cited the above-mentioned interesting work in our manuscript.
As a result, the Reviewer suggests reconsidering the article after major revision.
Round 2
Reviewer 1 Report
Dear Author's,
Thank you for addressing the concerns. Could you please expand on that why both gender's were not included? Thank you.
Author Response
Thank you for addressing the concerns. Could you please expand on that why both gender's were not included? Thank you.
- We thank the reviewer for this comment. Indeed, our study cohort was largely male and thus did not allow separate interpretation of the data according to gender. This might be attributable to the higher incidence and prevalence of HF in men that has been reported at all ages. We also mentioned this limitation in our limitation section with the following statement .
- Page 11, paragraph 2:
“In addition, we did not consider gender-specific differences in the calcium homeostasis as our study cohort was largely male, which, however, might be explained by the higher incidence and prevalence of HF in men that has been reported at all ages [42].“

Reviewer 4 Report
The manuscript by Kirsten Thiele et al. entitled “The role of vitamin D3 as independent predicting marker for one-year mortality in patients with acute heart failure” aimed to investigate whether vitamin D3 levels, assessed at hospital admission, are associated with survival at one-year in patients with acute decompensated HF. Specifically, the authors compared outcome estimates by serum levels of 25-OH-vitamin D3 and 1,25-(OH)2-vitamin D3 to those obtained from the well-established Seattle Heart Failure (SHF) Model.
The article is well written and leads some evidence to such point, however associations between levels of parathormone and one-year-mortality should be investigated.
